## Opinion: Distribute paleoscience information across the next IPCC reports

Darrell Kaufman[1] and  Valérie Masson-Delmotte[2]

[1] School of Earth & Sustainability, Northern Arizona University, Flagstaff AZ, USA

[2] Laboratoire des Sciences du Climat et de l'Environnement (UMR CEA-CNRS-UVSQ/IPSL 8212), Université Paris Saclay, France

Correspondence to: Darrell S. Kaufman (darrell.kaufman@nau.edu)

**Abstract.** In this opinion piece, we evaluate two approaches for incorporating paleoscience information into future assessment reports by the Intergovernmental Panel on Climate Change (IPCC). One approach advocates for a dedicated paleoclimate chapter, while the other supports the continued integration of paleoscience with other lines of evidence across multiple sections of the report, as done in the most recent assessment cycle. We address the merits and challenges of these two approaches. We argue that paleoscience expertise is most effectively deployed where it leads to integration of paleoscience knowledge and demonstration of its policy relevance, and we suggest opportunities for expanding paleoscience contributions in future IPCC reports, regardless of the approach chosen.

### Introduction

As the scoping phase for the next IPCC report is underway (IPCC, 2024a), discussions within the paleoscience community have emerged regarding the most effective way to incorporate paleoscience information into future reports. Some advocate for the inclusion of a dedicated chapter on paleoclimate (Esper et al., 2024; PAGES, 2024), as was done in previous IPCC reports. Proponents of this approach contend that a separate chapter would provide a more comprehensive assessment of paleoclimate information and ensure the visibility of this important field. They argue that this approach could safeguard the representation of paleoclimate experts as IPCC authors, together with increased relevancy and visibility of paleoscience (PAGES, 2024). In contrast, we argue here that paleoscience is made more relevant to the target audience – those interested in current and future climate change, risks and responses – and that it is afforded greater visibility when the information is distributed and integrated with other lines of evidence across the reports, as it was in the latest IPCC Assessment Reports (AR6). In this distributed approach, knowledge of pre-instrumental, pre-industrial climate is considered alongside and on par with other multiple lines of evidence

including observations, theory and modeling that are needed for a robust and comprehensive assessment of the state of knowledge, including the assignment of confidence levels. We find
that insights from paleoscience were promoted in AR6 because, rather than consolidating the subject within a separate chapter where it might appear unconnected to actionable knowledge,
the relevance of pre-industrial climate change was highlighted in multiple chapters of the Working Group I report (WGI; IPCC, 2021a) where current and projected climate changes were
placed into a broader context of long-term natural variability.

        The outline for AR6 Working Group I report (IPCC, 2021a) was scoped following

extensive discussions by an international group of select climate experts, with input from the broader community (IPCC, 2018a), and guided by a vision for a holistic and integrative report
(IPCC, 2018b). The resulting outline focused on the state of the climate system, processes that shape global and regional climate responses, and regional information (section 1.1.2 of Chen et
al. (2021) explains the rationale for the AR6-WGI structure and its relation to the previous AR5-WGI report). In AR6, paleoscience content was further distributed in the Working Group II report
(IPCC, 2022a) as it relates to the detection and attribution of ecosystem changes, and the vulnerability and adaptation of Earth's biota, socio-ecological systems and societies to past
climate variations (Cross-Chapter Box PALEO, Vulnerability and Adaptation to Past Climate Change, in Ara Begum et al., 2022). And in the Working Group III report (IPCC, 2022b),
centennial and longer timescales were part of the assessment of carbon storage and removal.

        While the previous two assessment reports (AR4 and AR5) had included a separate

chapter focused on paleoclimate, this was not the case for AR6. The separate chapters in AR4 and AR5 did much to advance the assessment of the state of knowledge from paleoclimate
archives in those reports; however, considering the purpose of IPCC reports along with advances in paleoscience, we see distinct advantages to the distributed approach adopted for
AR6, where paleoscience information was integrated with other lines of evidence whenever possible based on available literature.
We view the expansion of paleoscience information across AR6 as integral to the maturation of scientific knowledge. While maturation focused on specific past periods builds
depth and specialization, integration across lines of evidence and timescales enables a more holistic understanding of such complex phenomena as the response of the Earth system to
natural and anthropogenic forcings. In AR6 for instance, proxy-based reconstructions provided a long-term perspective on the evolution of modes of variability (Cassou et al., 2021). This
integration of knowledge across fields of climate science is facilitated by the IPCC assessment process, which strengthens interactions among scientists with complementary expertise (Weart,
2013). It enhances the robustness and relevance of knowledge, making it a more powerful and comprehensive process. This holistic approach ultimately accelerates the maturation of
knowledge by fostering a more interconnected, accurate, and actionable understanding of climate science.

**The case for a separate chapter and the benefits of the distributed approach**

A separate chapter has been proposed as a means to facilitate a more complete assessment of paleoscience information. Thoroughness is indeed a core principle of IPCC assessments, but
there are practical limits to what can be included and the reports have already been criticized for being too sprawling. The study of past Earth system changes is a huge field and the diversity of
scientists selected as IPCC authors must encompass its full breadth of expertise. Considering the exponential rise in climate science evidence from the literature (Masson-Delmotte, 2024)
and faced with very tight constraints on the number of words and pages available for any one topic, we submit that the limited paleoscience information that is included is most effectively
deployed where it leads to integration of paleoscience knowledge and demonstration of its relevance. The distributed approach facilities a more complete assessment by avoiding potential
gaps where paleoscience information might have contributed to informed decision-making. This relevance dimension is core to IPCC assessments, which thus differ from textbooks or in-depth
reviews for specialized audiences.

While some have proposed both a dedicated chapter and distributed information, this

presents practical challenges in terms of author selection and maintaining consistency across chapters. Considering all of the other dimensions of climate science expertise to represent when
selecting IPCC authors, the fair share of paleoscience experts among the group of IPCC Lead Authors would be too small to both populate a separate paleoclimate chapter and to embed into
other chapters. Embedding paleoscience authors within each chapter team is needed to ensure that the paleo perspective is effectively included within the context of those topics and their
high-level, policy-relevant findings of the type that are promoted to chapter Executive Summaries, which underpin the summary documents. Moreover, this would increase the
challenges to ensure consistency and complementarity among chapters and reports, and to avoid gaps. We believe that the involvement of the limited number of paleoscience authors in
IPCC reports is particularly crucial for ensuring the integration of paleoscience knowledge wherever possible, thereby demonstrating policy-relevant outcomes to a broader audience.
A separate chapter is also seen as a platform for a team of experts to work together closely to assess topics in more depth and deliver a more robust and detailed assessment of
uncertainties, compared to the distributed approach. Whether information from paleoscience comprises a separate chapter or is distributed across chapters, the quality in IPCC reports is
upheld through an extensive open review process overseen by designated Review Editors. These subject-matter experts ensure that all substantive comments are addressed in a
balanced and transparent way. In our experience, and from our conversations with other IPCC authors, the content of the reports are more thoroughly reviewed and heavily scrutinized than
any single publication in peer-reviewed journals. The quality of the information in IPCC reports can also be attributed to the readily accessible data that underlie the major findings, which
enables traceability and reproducibility.

           A separate chapter focusing on paleoscience could make it easier to locate information

about the subject. However, the field of climate science is far too large and rapidly growing for each discipline in Earth system observations, theory, processes and projections to have their
own convenient chapter. Instead, in AR6-WGI, key paleoscience information from across the chapters came together in a dedicated box in the Technical Summary (Box TS.2,
"Paleoclimate," in Arias et al., 2021) as part of the report's distillation process in support of the Summary for Policy Makers. A cross-chapter box in AR6-WGI-Chapter 2 (Changing State of the
Climate System; Gulev et al., 2021) points to sections across the report that present information about each of multiple "paleoclimate reference periods," periods that have been extensively
studied based on both empirical evidence and climate modeling as examples of distinct climate states. Meanwhile, emerging artificial intelligence tools (e.g., Climate Q&A, 2024) offer new
user-friendly opportunities to interact with IPCC reports across individual chapters.

           Some see a separate chapter as providing greater visibility to paleoscience. We place

high value on visibility for raising awareness of our science across a broader audience, an opportunity afforded by the widely distributed IPCC reports. We emphasize that paleoscience
information is made more visible when it is covered more comprehensively, as we contend that it was in AR6-WGI than in previous IPCC reports. This is evidenced by the breadth of topics
informed by paleoscience information across the report, including the summary documents (PAGES, 2022; Masson-Delmotte, 2021), and by a textural analysis of its content (see below).
Some of this expanded coverage reflects paleoscience knowledge developments, with longer time span between reports and more material to be assessed. Plus, paleoscience chapters in
AR4 and AR5 had themselves stimulated new paleoscience research.
**Paleoscience coverage in AR6-WGI compared with previous reports**

Despite the absence of a chapter dedicated to paleoscience, or because of the choice of more

holistic approach for the report structure designed to integrate multiple lines of evidence,
       paleoscience information was featured more prominently in AR6 than in the previous two

reports (AR5 and AR4). This assertion is based on an analysis of the contents of the WGI
       Summary for Policy Makers (SPM) (IPCC, 2021b), which highlights findings of greatest

relevance to decision makers. Specifically, the keywords "paleo" or "millennia"(l), which were
       typically used when assessing pre-industrial climate at multiple time scales, were mentioned

more frequently in AR6-WGI-SPM than in those of AR5 (IPCC, 2013) and AR4 (IPCC, 2007),
       both as a total number of occurrences and relative to the number of pages (Table 1). The

frequency of key findings based on paleoscience evidence were also greater in AR6-WGI-SPM,
       as was the number of words that comprise these findings (Table 1). Our simple keyword search

leads to the same conclusion as that of one of the preprint referees for this manuscript (Lunt,
       2024) who independently surveyed the two SPMs for mentions of paleoscience information.

150           Another preprint referee (Brierley, 2024a) surveyed the frequency of citations to *Climate
       of the Past*. He found that this journal was cited 122 times in AR5-WGI versus 163 times in

AR6-WGI. This increase in citations represents an approximately constant proportion of the total
       number of works cited in the WGI contributions to AR5 and AR6 (1.34% vs 1.25%, respectively).

However, this metric needs to be seen in context of the huge expansion of papers published
       across the field of climate change generally. The number of peer-reviewed papers with the

keyword "climate change" published in the year the AR5-WGI report was released was one-third
       the number for the AR6-WGI report (approximately 5000 in 2013 versus 15,000 in 2021 based

on Web of Science accessed September 2024). This compares with the number of papers
       published per year by this journal, which increased by one third (130 in 2013 versus 173 in

2021). Therefore, the importance of paleoscience as represented by the proportion of *Climate of
       the Past* citations compared to all other citations in the WGI report was essentially equal

between AR5 and AR6, despite the huge growth of climate publications overall (300%)
       compared with the modest growth of *Climate of the Past* publications during the same period

(33%). This analysis addresses the extent to which paleoscience was considered across the
       WGI reports of AR5 and AR6 rather than their SPMs alone, and it supports our contention that

paleoscience was featured more prominently in AR6.
              Paleoscience has been a part of IPCC reports since the beginning, and the SPMs of all

previous WGI reports contain findings that attest to increasingly unprecedented changes in the
       climate system over centuries and millennia. The latest AR6-WGI-SPM expands on these

findings by describing evidence from additional indicators of the state of the climate system

beyond atmospheric greenhouse gas concentrations and large-scale surface temperature.

Paleoscience in the AR6-WGI-SPM looks further back in time to climate states with higher global warming levels than in previous assessment reports. It is mentioned along with other
evidence that narrows the uncertainty range of climate sensitivity and strengthens confidence in projections of long-term sea-level responses to different levels of sustained warming. It is also
used to evaluate low-likelihood events with high-impact outcomes, including large explosive volcanic eruptions and their documented climate effects.
A similar expansion of paleoscience information is also seen in the Technical Summary (TS) of the AR6-WGI report (Arias et al., 2021) compared with the previous two reports (Stocker
et al., 2013; Solomon et al., 2007). All three contain a section or box dedicated to paleoscience. Outside of these more specialized sections, the keywords "paleo" or "millennia"(l) are mentioned
more frequently in AR6-WGI-TS than in AR4 and AR5 relative to their number of pages (Table 1). Furthermore, AR6-WGI-TS includes seven figures that feature paleoscience information
compared with five in AR4 and three in AR5. Among these figures is a direct comparison of atmospheric carbon dioxide levels back through the Cenozoic and forward through alternative
projections to 2300, including both timeseries and maps of global temperature. This is the first time a figure with these global-scale climate indicators has appeared in an IPCC report; it is
indicative of the integrative approach in AR6, with attention to placing current and projected changes into a long-term context. Finally, to reach a broader audience, paleoscience
information is included in more Frequently Asked Questions of AR6 (IPCC, 2021c) than in previous reports (Table 1), and simulations of paleoclimate reference periods are incorporated
into the Interactive Atlas alongside historical runs and climate projections from the same models (Gutiérrez et al., 2021).

**Challenges of the distributed approach**

While we view the integration of paleoscience topics throughout the IPCC report as an inevitable and healthy progression for an increasingly expansive and relevant subject, we are
fully aware of its challenges. When writing the reports, paleoscience authors need to coordinate closely to avoid redundancies, ensure important topics do not fall into cracks between chapters,
and prepare dedicated cross-chapter boxes as entry points for paleoscience topics. For example, cross-chapter boxes in AR6-WGI Chapter 2 (Gulev et al., 2021) focuses on multiple
"paleoclimate reference periods" and another features the climate of the Pliocene when $CO_2$ concentrations were last similar to those of present day. Box TS.2 (Arias et al., 2021) includes a
synthesis of the assessed values for global mean temperature, atmospheric carbon dioxide, and

global mean sea level for multiple paleoclimate reference periods, and directly compares global

mean temperatures derived from observations with those from climate models for these
      reference periods, with all of the data accessible and traceable.

208         These boxes were written by AR6 paleoscience Lead Authors and Contributing Authors
      from multiple chapters who formed one of several WGI breakout groups that focused on cross-

cutting topics. Paleoscience authors of future reports should be prepared to devote additional
      time to serve their roles as authors within both their chapter teams and the paleo breakout

group. In the future, more formal coordination mechanisms could include a new role for cross-
      chapter paleoscience coordinators who could participate in Coordinating Lead Author meetings

and work proactively across the Working Groups to assure insights from paleo evidence are
      considered within chapters where decisions about the specific content are made, as guided by

the scoping document. This way, novel and relevant paleoscience findings are more likely to be
      promoted to the Executive Summary of each chapter, which underpins the most widely read

summary documents (TS and SPM). The author team for these summary documents needs to
      include paleoscientists who can draw together key findings disseminated across chapters.

220         Despite their increasing length, space available for any climate change subject is
      exceedingly limited in IPCC reports. There is generally no scope nor purpose for extensive

analyses of datasets, textbook-style reviews of methods or lengthy discussions of knowledge
      gaps aimed at experts. Instead, IPCC reports rely heavily on evidence from timely published

literature, including community-based assessments of relevant topics that provide in-depth
      analysis of methodologies, outcomes and uncertainties, and that support the integration and

distillation of information within IPCC assessments of policy-relevant information.
**Opportunities for future reports**
      Regardless of whether the information is consolidated in a separate chapter or distributed

through the report, the success of paleoscience in future reports depends primarily on
      community efforts to advance the state of knowledge and evaluate uncertainties within timely

academic publications. It also depends on input from paleoscientists during the scoping phase
      of the report (planned in December 2024 for AR7) so the full breadth of relevant paleoscience

topics is explicitly identified and effectively parsed among chapters, and key expertise is
      ensured within the selection of author teams. Timely publications calling for specific topics to be

addressed, with suggestions for scoping are also valuable. Individuals and organizations have
      input to this process through their appointed IPCC National Focal Points and Observer

Organizations (IPCC, 2024b). They can advocate for topics and keywords to be included in

chapter outlines or cross-chapter boxes, which will help ensure that the author selection

includes the right balance of expertise.

Like the PAGES communication that motivated this piece (PAGES, 2024), we too

encourage paleoscientists to support and engage in the IPCC process. Among the various

avenues for participation (IPCC, 2024c) is volunteering as a reviewer during the drafting and

revision phases to make sure that new knowledge developments are included where relevant.

Collective reviews of IPCC reports by early career scientists can be especially fruitful, as it was

for AR6, and this activity could be strengthened for future reports (Moreno-Ibáñez et al., 2024).

In addition, Contributing Authors play an important role as content experts to help draft chapter

text alongside Lead Authors. In Chapter 2 of AR6-WGI (Gulev et al., 2021), for example, 22

paleoscientists served as Contributing Authors from outside the WGI Lead Author team.

Paleoscience, like all climate science communities competing for coverage in this high-

level product, can work proactively and in concert with the current IPCC assessment cycle to

generate or update systematic reviews of the state of knowledge regarding understanding past

climate variations and their implications, and regarding key policy-relevant topics. Now is the

time to identify what appraisals of major research advances that address socially relevant

understanding of climate change (e.g., Kaufman, 2020) are missing from the literature and to

initiate coordinated efforts by experts to fill these gaps in support of AR7. Paleontologists too

have opportunities to expand their contribution to providing policy-relevant information on

climate change impacts (Kiessling et al., 2023). An example of such a community-led effort in

support of a key IPCC topic is that by the World Climate Research Programme (WCRP) for the

grand challenge of understanding climate sensitivity (Sherwood et al., 2020), with new

developments underway to inform AR7 (e.g., Cooper et al., 2024).

We see potential for stronger inclusion of information regarding topics such as climate

extremes of the past, implications of different durations of different sustained levels of warming,

past abrupt events, irreversibility, and insights related to the vulnerability and adaptation of

ecosystems and biodiversity. Major collective efforts are needed to evaluate and communicate

the state of understanding of past climate variations at global to regional levels.

While the AR6 placed a stronger emphasis on regional climate information than previous

reports, advances are needed to include paleoscience information in the distillation of regionally

relevant climate information. This includes, for example, regional-scale seasonal and annual

hydroclimate reconstructions, extreme events and climatic impact-drivers. Syntheses based on

transparent approaches and supported by well-curated and readily traceable data are especially

useful. This includes updates of paleoclimate forcings and of key indicators of the state of global

climate and their uncertainties – including the limitations of paleo data assimilation products –

for well-studied paleoclimate reference periods. Considering the emphasis on climate modeling

in IPCC reports, efforts directed toward model evaluation and other CMIP7 (2024) and PMIP5

(Brierley, 2024b) science goals are crucial. We see the need for expanded use of evidence from

paleoscience for assessing climate model fitness-for-purpose and confidence in projections

grounded in rigorous model-data comparisons, especially for the paleoclimate reference

periods, and where there is deep uncertainty, including for instance tipping points, Antarctic sea

ice, or land carbon feedbacks.

       In addition to their core mandate, IPCC reports also contribute to strengthening climate

literacy. Report elements designed for schoolteachers and the general public include

"Frequently Asked Questions," which address key topics with up-to-date information in a

consistent style and have been bundled into a single pdf (Connors et al., 2022). New for AR6 is

the colorfully illustrated, plain language "Summary for All" (IPCC, 2022c), which is translated

into multiple languages. Considering the widespread misconceptions and outdated views of past

climate variations, we see a need to distillate the current state of knowledge using accessible

plain-language text and scientifically rigorous, user-friendly data visualizations, anchored in a

co-design process (Morelli et al., 2021; InfoDesignLab, 2024). We argue that paleoclimate

literacy can be strengthened by clear communication of topics such as the causes, mechanisms

and characteristics of past climate changes, lessons from past climates that are relevant for

well-informed climate action, and how recent and future changes compare with those of the

past. This includes improving the display of post-industrial changes in key climate system

indicators, such as global mean surface temperature, in context of long-term changes in a way

that the general public and decision-makers can easily understand.

296            The underlying publications with these advances are needed in support of the AR7

assessment cycle. These advances will require substantial support for community efforts, both

by funding agencies and by professional organizations equipped for regional and international

coordination. Considering the major expansion of paleoscience knowledge since AR6, such

products are beyond what can be produced by a small group of IPCC authors regardless of

whether those authors work within a single chapter or are distributed across the Working

Groups.
**Competing interests**

The authors were involved in the preparation of AR6.


## Acknowledgement

We thank M. Grosjean, F. Lambert, N. McKay, T. Stocker, and P. Thorne for their valued input on an earlier version of the manuscript, although not all fully shared our views. We thank the two
referees (C. Brierley, D. Lunt) and members of the paleoclimate community (N. Abram, K. Allen, K. Anchukaitis, E. Arellano-Torres, I. Cacho, C. Martin-Puertas, M. Prather, M. Sigl, J. Smerdon,
A. Voelker, R. Wilson, E. Wolff) for their comments and suggestions during the open discussion phase of the manuscript review. C. Brierley provided the count of citations to articles in *Climate*
*of the Past*.

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

Table 1. The frequency of paleoscience information in the Working Group I contributions to the
last three IPCC climate assessment reports.

| Working Group I | AR4 | AR5 | AR6 |
|---|---|---|---|
| Publication year | 2007 | 2014 | 2021 |
| Paleoclimate chapter in report | Yes | Yes | No |
| FAQs with paleo content | 2 | 2 | 4 |
| CP citations | NA | 123 | 175 |
| | | | |
| Summary for Policy Makers | | | |
| Total pages of content | 17 | 26 | 28 |
| "Paleo" or "millennia" mentions* | 6 | 8 | 15 |
| Average mentions per page | 0.35 | 0.31 | 0.54 |
| Major sections with paleo content | 3 | 2 | 3 |
| Bullets/subsections with paleo content | 4 | 6 | 8 |
| Approx. words containing paleo content | 390 | 360 | 460 |
| Figures with paleo content | 1 | 0 | 1 |
| | | | |
| Technical Summary | | | |
| Total pages of content | 71 | 82 | 107 |
| "Paleo" or "millennia" mentions** | 19 | 38 | 56 |
| Average mentions per page | 0.27 | 0.46 | 0.52 |
| Figures with paleo content | 5 | 3 | 7 |

* Includes "palaeo" and "millennial"

** Not counting the paleo box or paleo perspective or text within figures
and their captions
