# Peer review of "Opinion: Distribute paleoscience information across the next IPCC reports"

_EGUsphere, 2024_

## Referee Comment (RC2)

I thought that it would be useful to carry out this review in 3 stages:

(1) Independent thoughts on the topic based on the title of the paper alone, before reading the paper.
(2) A review of the paper itself before reading any online comments.
(3) Any additional thoughts after reading all online comments.

I should also declare an "interest" in that I was an IPCC AR6 Lead Author (on Chapter 7, The Earth's Energy Budget, Climate Feedbacks and Climate Sensitivity).

**(1) Independent thoughts**

My personal experience being involved in AR6 was that "embedding" paleoclimate science in individual chapters, rather than having a separate paleoclimate chapter, had a number of effects:

(a) My belief is that the non-paleo Lead Authors (on the Chapter that I was involved in) were likely more aware of the paleo content of the report than they would have been if paleo was a separate chapter.  As such, I expect that the paleoclimate information was better integrated into the non-paleo science of my chapter, and fed in more to overall assessment statements.  A similar effect likely occurred in other Chapters.
(b) A result of (a) was, I believe, that paleoclimate information ended up being more prominent in the Summary for Policymakers than it would have been otherwise.  For example, there is a paleo figure in the SPM of AR6 but not in AR5.  There are 8 specific mentions of paleoclimate information or time periods in the AR6 SPM (A.2, A.2.1, A.2.2, A.2.3, A.2.4, A.4, B.1.1, C.1.4), and 4 in the AR5 APM (B.1.4, B.4.4, B.5.2, E.7.1).
(c) The lack of a paleoclimate chapter in AR6 makes it less useful as a "textbook" for paleoclimate information than AR5.
(d) Although there were paleoclimate scientists as Lead Authors in several AR6 chapters, there was not one in Chapter 3, in which much model evaluation was carried out.  In my view, this was a loss compared to AR5, when there was a paleoclimate scientists as Lead Author on the model evaluation chapter (Chapter 9).  My impression is that, as a result of this, the use of paleoclimate data in model evaluation in AR6 was under-used (this is, in fact, an argument for distributing the paleo information across Chapters, but ensuring that is supported with paleo Lead Authors in the key Chapters).

Overall, it is my belief that paleoclimate information was better integrated into the AR6 report than into the AR5 report (although see (d) above), and as a result was more prominent in the SPM, and as a result likely better informed policymakers (which is, of course, the ultimate aim of IPCC).

**(2) Review of the paper**

Overall, I think that this paper gives a good summary of some of the arguments in favour of distributing the paleoclimate information across chapters in AR7, as was done in AR6.  However, I think that some more space could be dedicated to the opposing viewpoint, and summaries given of the main arguments in favour of the AR5 approach.  In addition, some of the statements could be better evidenced.

The evidence presented focuses on the inclusion of paleoclimate information in the SPM and Technical Summary.  The main purpose of IPCC is to inform policymakers, so I was wondering if this aspect can also be assessed?  For example, is there evidence that there were more governmental review comments on the draft report on paleo sections in AR6 compared with AR5?  Or more time spent on paleo aspects in the government approval plenary?

I'd also be interested to know how the amount of paleo information across the whole report differed in AR6 compared with AR5.  This would be easy to calculate for AR5, but would require a bit of work for AR6, to add up the content across multiple chapters, although the "paleo contents page" in Chapter 2, and/or the PAGES (2021) article would be helpful for this.  On this note, line 32-34: PAGES (2021) is given as a reference for "paleoscience information was covered more comprehensively in AR6-WGI than in previous IPCC reports", but that statement is not evidenced in that PAGES article, which is just a series of links to the AR6 paleo sections.

Line 35-36: It is an interesting claim that having a separate chapter stimulated more paleo research – is there any evidence for this?

Lines 59-62: Link to the Appendix which evidences this.

Line 85: Link to the Appendix which gives the exact numbers of FAQs.

**(3) Overview of online comments (as of 19/7/2024)**

The online comments and review provide a range of views and many interesting and important points, and I would encourage the authors to incorporate these into their revised paper.

CC2: It is an interesting suggestion to include BOTH a paleoclimate chapter and integration into individual chapters.  However, I feel that the integration requires paleo Lead Authors in individual chapters, and if you add to this the paleo chapter Lead Authors, this will be unfeasible in terms of the number of paleo scientists in the report compared to other disciplines.

CC6: I fully agree with all aspects of this comment, which I think supports my own independent review above.

RC1: I also fully agree with the comments in this review.  However, seeing that it was another paleo scientist involved directly in AR6, and also a modeler like me, I am slightly uncomfortable with the potential biases of the reviewers; however, I think that the wide engagement of the community in commenting on the article does go some way to counter that bias.

---

## Author Response (AR3)

Oct 22, 2024

Dear Qiong:
Thank you for your careful review of our revised manuscript and for your suggestions for improving the presentation. We have made almost all of the revisions that you suggested and have explained the rationale for those that we did not adopt. We believe that the revisions present a more balanced representation of the advantages and disadvantages of the two approaches.

Thank you.
Darrell and Valérie
* * *
Editor's suggestions in plain font, and authors' response in red font.

1. For the section headline "Response to arguments favoring a separate chapter."
The current headline comes across as somewhat argumentative, which might be at odds with the neutral and objective tone of a scientific manuscript. To enhance clarity and maintain a formal tone, I recommend considering a more neutral headline, such as: "Consideration of proposals for a separate chapter", "Evaluation of the case for a dedicated paleoclimate chapter". These alternatives keep the focus on the evaluation of the subject without suggesting a rebuttal.

Heading revised as suggested. Also added "and the benefits of the distributed approach" because this section includes both perspectives.

Some text in this new section contain a relatively strong argumentative tone. The wording often emphasizes direct rebuttals to the idea of a separate paleoclimate chapter, which might be seen as confrontational or overly defensive in a scientific publication. While it's important to present counterarguments, softening the language can create a more balanced and constructive tone.

Line 71-72: "Some see a separate chapter as a means to a more complete assessment of paleoscience information."
Suggested revision: "A separate chapter has been proposed as a means to facilitate a more complete assessment of paleoscience information. While this approach has advantages, there are also limitations to consider, such as the risk of isolating paleoscience from the broader context of the climate assessment."

First sentence revised as suggested. However, we did not insert the suggested second sentence because: (1) The topic of this paragraph is specifically the "completeness/thoroughness" of the assessment; the second sentence should follow the paragraph topic. (2) The point about risking isolation is already made earlier, on lines 31-33.

Line 78-80: "We argue that the limited paleoscience information that is included is most effectively deployed where it leads to integration..."
Suggested revision: "In our view, the integration of paleoscience information within other chapters offers a more effective way to demonstrate its relevance..."

Omitted and replaced the word "argue" as suggested. However, we did not remove the phrase about "the limited paleoscience information that is included..." because the point of this sentence and the paragraph is specifically the limited allotment of space available in the reports.

Line 84-86: "Some have suggested that paleoscience information should be included both in a dedicated chapter plus distributed in other chapters. However, this would be difficult to achieve in practice..."
Suggested revision: "While some have proposed both a dedicated chapter and distributed information, this presents practical challenges in terms of author selection and maintaining consistency across chapters..."

Text revised as suggested.

Line 94-95: "We argue that the contribution of the handful of paleoscience authors in IPCC reports is most critical for the integration of advances in paleoscience knowledge..."
Suggested revision: "We believe that the involvement of paleoscience authors in IPCC reports is particularly crucial for ensuring the integration of paleoscience knowledge across relevant chapters."

Text revised as suggested but retained "limited number of authors" because we want to keep the focus on the fact that there will be too few paleo authors.

2. The Introduction is comprehensive, but it jumps between contrasting viewpoints (integration vs. separation of paleoclimate in IPCC reports). The Introduction begins by briefly mentioning the scoping phase of the next IPCC report and quickly jumps into the debate between proponents of a separate paleoclimate chapter and those favoring integration (Lines 20–26).

Revised the first paragraph to progress more logically, as suggested, rather than jumping between contrasting viewpoints.

It would be smoother by introducing the overarching issue without immediately jumping into contrasting viewpoints. For example:
"As the scoping phase for the next IPCC report (IPCC, 2024a) is underway, discussions within the paleoclimate community have emerged regarding the most effective way to incorporate paleoclimate information into future reports. The role of paleoscience in providing long-term perspectives on climate change is crucial for understanding current and future climate variability."

First sentence revised as suggested. The second sentence is common knowledge among *Climate of the Past* readers and would add to already long first paragraph.

Then begins presenting the rationale for a separate paleoclimate chapter in a neutral tone, ensuring it is given a fair explanation. For example:
"Some in the paleoscience community suggest for the inclusion of a dedicated chapter on paleoclimate, as was done in previous IPCC reports (Esper et al., 2024; PAGES, 2024). Proponents of this approach claim that a separate chapter would provide a more comprehensive assessment of paleoclimate data and ensure the visibility of this important field. They argue that this approach could safeguard the representation of paleoclimate experts and emphasize the relevance of long-term climate variability in the context of modern changes."

Revised text as suggested.

After this, the authors could discuss the case for distributing paleoclimate information across multiple chapters, clearly contrasting it with the previous argument. Followed by transition into the authors' preferred approach. Then provide a brief summary of why authors believe the distributed approach is more beneficial, linking it to the overall goals of the IPCC reports.

The restructure separates the contrasting viewpoints, allowing readers to understand each perspective before moving into the authors' preference, making it easier for readers to follow the arguments.

Revised the first paragraph to progress more logically, as suggested, rather than jumping between contrasting viewpoints.

3. Abstract. For a more neutral and objective tone, the abstract should avoid the impression of a rebuttal and present the discussion as a balanced evaluation of the two approaches. Here is a suggestion to revise the abstract:
"In preparing for the 7th assessment reports by the Intergovernmental Panel on Climate Change (IPCC), the paleoscience community faces a decision on how best to present paleoclimate information. Two approaches are being considered: a dedicated paleoclimate chapter or the integration of paleoclimate data across multiple chapters. This manuscript evaluates both options, considering the potential benefits and challenges of each. While a dedicated chapter could enhance visibility and focus, integrating paleoclimate information throughout the report may provide broader context and relevance. Based on this evaluation, we suggest opportunities for improving paleoscience contributions in future IPCC reports, regardless of the approach chosen."

Revised the abstract essentially as suggested by presenting the discussion as a balanced evaluation of the two approaches. However, to say that "the paleoscience community

faces a decision" is misleading. The decision is in the hands of the IPCC. Also, to say that "a dedicated chapter could enhance visibility…" is contrary to our primary argument and would be confusing to pose it in the abstract.

3. Suggestion on overall structure
• Introduction
• Evaluation of the case for a dedicated paleoclimate chapter (change the headline from previous "Response to arguments favoring a separate chapter")
• Paleoscience coverage in AR6-WGI compared with previous reports
• Benifits and challenges of the distributed approach (merge the "Challenges" section with benefits from previous sections)
• Opportunities for future reports (more specific for the opportunities)

Revised the second and fifth headings as suggested. Moving the discussion of benefits of the distributed approach into the fourth section along with its challenges would be difficult because the benefits are the counterarguments in response to the case for the separate chapter, which is in the second section. Instead, we revised the heading for the second section to alert readers to location of this content.
* * *
October 8, 2024

Dear Dr. Zhang:

We have completed the revisions to our manuscript, "Distribute paleoscience information across the next IPCC reports." The revised version includes changes that were suggested by the two referees and by the community commenters. We already wrote a detailed point-by-point reply to each of the comments as part of the interactive discussion available on the EGUSphere preprint site. The revisions are shown as tracked changes to the original text in the uploaded file. The revisions include:

**Opinion piece.** We changed "Rapid Communication" to "Opinion" in the title, as requested. We also added "in this opinion piece, we…" to the abstract.

**More balanced presentation.** We added a lengthy section with the heading, "response to arguments favoring a separate chapter" to expand our discussion of the opposing viewpoint. The section considers the issues presented by the reviewers who commented on accuracy, completeness, visibility and related topics.

**Opportunities for new directions.** We expanded on the specific information from paleoscience that we see as having potential for stronger inclusion in future reports. For example, we point to the need for expanded use of evidence from paleoscience for assessing model fitness for purpose and confidence in projections. We also highlight an example of such a community-led effort in support of a key IPCC topic.

**Avenues for input.** We now discuss avenues for participation in addition to those already included in the original version.

**Validity of the textural analysis.** Our manuscript explains that our keyword search leads to the same conclusion as that of one of the preprint referees who surveyed the two SPMs for mentions of paleoclimate information. In addition, we added the frequency of citations to this journal, as provided by the second referee, to the discussion and to Table 1.

**Benefits of integration.** We added text to the revised manuscript to strengthen the point about multiple lines of evidence that must come together in support of assigning confidence levels to high-level, policy-relevant climate science conclusions.

**Impracticality of both a dedicated chapter and distribution.** We added a paragraph to the revised manuscript to address this suggestion by explaining that there are too few paleoscience Lead Authors to cover both bases and reiterating our view that paleoscience expertise in AR7 would be most effectively deployed where it leads to integration of paleoscience knowledge and demonstration of its relevance.

**AR6 author procedures.** We added some sentences to expand on the process that AR6 paleoscience authors used to help coordinate their efforts. This includes the important role played by Contributing Authors.

**Additional reference.** We now cite the Esper et al. (2024) perspective piece, which called for a separate paleoclimate chapter in AR7.

**Link to climate modeling.** We called out the connection with CMIP7 and PMIP 5.

**Paleo topics covered in AR6.** We now state that it is our opinion that the coverage was greater in AR6 as evidenced by the breadth of topics that considered paleoscience information as listed in the table of contents hosted on the PAGES website.

**Table 1.** We now include a Table instead of an Appendix and we call it out more frequently.

**References.** We adjusted the style to the one used by *Climate of the Past*.
* * *
Referee's verbatim comments are in plain font, followed by authors' replies in **bold**.

==**Referee 1, Chris Brierley**==
This rapid communication tackles an important question for our field: "how can we make paleoclimate research societally relevant?" In my opinion, it warrants publication in

Climate of the Past. I have a couple suggestions, which the authors may want to consider prior to publication.

**We thank the referee for his insights and suggestions, and for clearly recognizing the overall purpose of our piece.**

If I was writing it, I might have taken a more mollifying tone. It could have started with the idealised position of Dr Arellano-Torres (dedicated chapter and distributed throughout, https://doi.org/10.5194/egusphere-2024-1845-CC2); said that was unrealistic; presented the evaluation of the change in approach taken by AR6; and finally proposed continuation of that approach. But I didn't write it - or even think to do so. The authors' more forthright narrative is also an acceptable narrative.

**In response to this suggestion, and to those of other Community Commenters, we will include a separate section on arguments favoring a separate chapter.**

I had not interpreted the plea for engagement by PAGES (2024) solely as a request to push for dedicated paleoscience chapter. One could alternately interpret it as worry that paleoscience might be sidelined in the upcoming AR7. I feel that this worry about AR7 is legitimate, and our community runs that risk if we don't engage with its scoping and writing process. It is even mentioned on L92. In PAGES (2024), this worry was amplified by a perception that paleoscience suffered from reduced visibility in the AR6. Your analysis tries to address that perception directly.

**We understand the concern that paleoscience might be sidelined in AR7 and agree that the main point of the PAGES-IPO communication was a call for participation in the IPCC process. In our revised manuscript, we will expand the section that explains how the paleoclimate community can be proactive in the IPCC process for the next cycle so as to reiterate and reinforce the PAGES call to action. However, the PAGES communication leads with several sentences that clearly state the perceived advantage of a dedicated paleoclimate chapter in terms of its better "visibility and relevance of paleoscience". It's this comment that motivates our piece.**

L52-55. This section starts with a sentence which summarises the conclusion of analysis that has not yet been presented. Is this sentence even necessary? The textual analysis is the key evidence of this article and should not be relegated to an appendix.

**We think that a strong first sentence is useful here. We will re-label the appendix as a table, as suggested.**

L126: You might to include the CMIP7 science goals here. They're still under consultation, but could be cited with https://wcrp-cmip.org/cmip-panel-meet-to-advance-cmip7-and-ar7-fast-track/

**Thank you for this useful suggestion. We will add, "Considering the emphasis on climate modeling in IPCC reports, efforts directed toward model evaluation and other CMIP7 (2024) and PMIP5 (Brierley, 2024, https://pastglobalchanges.org/news/137793) science goals are crucial." Brierley's update includes a link to a recorded PMIP seminar with further discussion about the latest paleoclimate modeling plans, which is available at:  https://www.youtube.com/watch?v=WD8zzDGY5l8**

95-109: In this section, you describe the process by which IPCC authors coordinate and distribute paleoscience across the chapters. I know that this requires a substantial and dedicated effort, because I have seen elements of it in action. Scientists outside of the IPCC may not be aware of this effort (which is potentially documented in the peer-reviewed literature for the first time here). You want to also use this Rapid Communication to be transparent about this process, allowing those who do not look beneath the IPCC chapter titles to see the effort and benefit of embedding paleoscience evidence throughout.

**Thank you for pointing this out. We will add some sentences to expand on the process that AR6 paleoscience authors used to help coordinate their efforts.**

L130: There will be many readers of this article who have no influence on the IPCC AR7 structure, will not participate in its creation, and are not in a position to contribute to community syntheses. I wonder if you could provide such individuals with a recommendation or two for their own research and how they might join in the effort of demonstrating the relevance of paleoscience?

**Some researchers might find it useful to read how their paleo specialty is treated within the context of multidisciplinary chapters of the IPCC report. An index of paleo topics in the WGI report is at https://pastglobalchanges.org/news/quick-guide-paleoclimate-ipcc-ar6-2021-report. A new large language model trained on all of AR6 is also available to explore how and where specific topics are covered in the report (www.climateQA.com). In addition, professional organizations can offer opportunities to engage in such transdisciplinary studies as well as providing easy avenues to join in community syntheses. Finally, as suggested, our revised manuscript will expand on information from paleoscience that we see has potential for stronger inclusion in future reports.**

Finally, I was pondering other quantitative ways of exploring the quantity and importance of paleoscience in the IPCC Assessment Reports. Given this article has been submitted to Climate of the Past, I have counted how often its articles are cited in the various chapters. Climate of the Past was first published in 2009, so cannot feature in AR4. I include the results in the two tables below. As is argued in the present manuscript, the citations of Climate of the Past are more widely distributed amongst the various chapters of AR6. I was surprised to discover that Climate of Past occurs slightly more often as a proportion of the total WGI citations in AR5, rather than AR6. Obviously, this data should be treated with caution as it uses a single journal's citation to track a whole field. It is probably also worth

noting that there may also have been a slight trend in scope of the journal over the past 15 years, and that the type of articles cited could have changed. For example, all 3 citations in AR5 WGI Chapter 2 relate to the creation of instrumental datasets that do not extend prior to 1900, and so may not represent paleoscience.

**We appreciate the effort by the Referee to consider other metrics of the extent to which topics are represented across the entire WGI contribution to the IPCC report. The fact that *Climate of the Past* was cited 122 times in AR5-WGI versus 163 times in AR6-WGI supports our contention that paleoscience was not diminished in AR6. This increase in the number of citations represents a similar proportion of the total number of works cited in the WGI contributions to AR5 and AR6 (1.34% vs 1.25%, respectively). However, this metric needs to be viewed in context of the huge expansion of papers published across the field of climate change generally. The overall number of papers that refer to "climate change" or "global warming" more than doubled during the years that preceded each of the two assessment reports (10,811 vs 28,991 per year; De-Gol et al., 2023; 10.1038/s44168-023-00072-3). Within the time span of AR6 alone, the number of papers with the keyword "climate change" doubled from around 30,000 to more than 60,000 per year between 2015 and 2022 (Masson-Delmontte, 2024; 10.1371/journal.pclm.0000451). This compares with the number of papers published by this journal, which increased by 39% (88 vs 122 per year). Therefore, the importance of paleoscience as represented by the proportion of *Climate of the Past* citations compared to all other citations in the WGI report was essentially equal between AR5 and AR6 (1.34% vs 1.25%, respectively) despite the huge growth of climate publications overall (168%) compared with the modest growth of *Climate of the Past* publications during the same period (39%). In other words, citations to *Climate of the Past* more than kept pace between AR5 and AR6 relative to the massive increase in climate papers overall. Providing the Referee agrees, we will add this information to the revised manuscript and to Table 1. It attests to the extent to which paleoscience is represented across the entire WGI report rather than the summary documents alone. Plus, the information would likely be of interest to readers of this journal.**

Referee's verbatim comments are in plain font, followed by authors' replies in **bold**.

**Referee 2, Dan Lunt**

I thought that it would be useful to carry out this review in 3 stages: (1) Independent thoughts on the topic based on the title of the paper alone, before reading the paper. (2) A review of the paper itself before reading any online comments. (3) Any additional thoughts after reading all online comments. I should also declare an "interest" in that I was an IPCC AR6 Lead Author (on Chapter 7, The Earth's Energy Budget, Climate Feedbacks and Climate Sensitivity).

**We thank the referee for sharing his views as informed by his experience as an AR6 Lead Author.**

(1) Independent thoughts
My personal experience being involved in AR6 was that "embedding" paleoclimate science in individual chapters, rather than having a separate paleoclimate chapter, had a number of effects:

(a) My belief is that the non-paleo Lead Authors (on the Chapter that I was involved in) were likely more aware of the paleo content of the report than they would have been if paleo was a separate chapter. As such, I expect that the paleoclimate information was better integrated into the non-paleo science of my chapter, and fed in more to overall assessment statements. A similar effect likely occurred in other Chapters.

**We agree that spreading paleoscience authors across chapters raises awareness of our science among the broader group of IPCC lead authors. Our revised manuscript will strengthen the point about multiple lines of evidence that must come together in support of assigning confidence levels to high-level, policy-relevant climate science conclusions. Here, and in response to Community Commenter 12, we agree that arriving at the most robust key conclusions (SPM headline statements) means that the handful of authors that represent paleoscience should be spread across all chapter teams where they can effectively represent and deploy the paleo perspective.**

(b) A result of (a) was, I believe, that paleoclimate information ended up being more prominent in the Summary for Policymakers than it would have been otherwise. For example, there is a paleo figure in the SPM of AR6 but not in AR5. There are 8 specific mentions of paleoclimate information or time periods in the AR6 SPM (A.2, A.2.1, A.2.2, A.2.3, A.2.4, A.4, B.1.1, C.1.4), and 4 in the AR5 SPM (B.1.4, B.4.4, B.5.2, E.7.1).

**Thank you for this independent appraisal of the prominence of paleoscience information in the AR6 versus AR5 SPMs. It is entirely consistent with ours: both show a substantial increase in AR6. We will revise our manuscript to say that our keyword search leads to the same conclusion as that of one of the preprint Referees who surveyed the two SPMs for mentions of paleoclimate information.**

(c) The lack of a paleoclimate chapter in AR6 makes it less useful as a "textbook" for paleoclimate information than AR5.

**True. IPCC reports are generally written for decision makers and, as we note in our piece, are not meant to serve as textbooks, as emphasized by Community Commenters 3 and 12. That said, the FAQs cover topics that are of special interest to students and the general public. Information about paleoscience is included in more of the FAQs in AR6-WGI than in previous reports, a clear effect of the distributed approach considering that each chapter is only represented by a few FAQs each.**

(d) Although there were paleoclimate scientists as Lead Authors in several AR6 chapters, there was not one in Chapter 3, in which much model evaluation was carried out. In my

view, this was a loss compared to AR5, when there was a paleoclimate scientists as Lead Author on the model evaluation chapter (Chapter 9). My impression is that, as a result of this, the use of paleoclimate data in model evaluation in AR6 was under-used (this is, in fact, an argument for distributing the paleo information across Chapters, but ensuring that is supported with paleo Lead Authors in the key Chapters).

**We agree that information from paleoscience provides a powerful approach to model evaluation. Its impact would be diminished if it were carried out within the confines of a chapter relegated to paleo topics instead of discussed along with other evidence in context of topics such as climate projections. We will expand on this point in the revised manuscript section on "opportunities" by pointing out the need for expanded use of evidence from paleoscience for assessing model fitness-for-purpose and confidence in projections.**

Overall, it is my belief that paleoclimate information was better integrated into the AR6 report than into the AR5 report (although see (d) above), and as a result was more prominent in the SPM, and as a result likely better informed policymakers (which is, of course, the ultimate aim of IPCC).

**We agree. We wrote the piece in hopes that others would also see the benefits of an integrated approach, which they might have previously overlooked.**

(2) Review of the paper
Overall, I think that this paper gives a good summary of some of the arguments in favour of distributing the paleoclimate information across chapters in AR7, as was done in AR6. However, I think that some more space could be dedicated to the opposing viewpoint, and summaries given of the main arguments in favour of the AR5 approach. In addition, some of the statements could be better evidenced.

**In addition to the challenges of the distributed approach that we already describe, we will expand our discussion of the opposing viewpoint by adding a section on arguments in favor of a separate chapter. We will also address the suggestion that paleoscience information be featured in both a dedicated chapter and distributed in other chapters.**

The evidence presented focuses on the inclusion of paleoclimate information in the SPM and Technical Summary. The main purpose of IPCC is to inform policymakers, so I was wondering if this aspect can also be assessed? For example, is there evidence that there were more governmental review comments on the draft report on paleo sections in AR6 compared with AR5? Or more time spent on paleo aspects in the government approval plenary?

**Assessing the impact of IPCC reports on policymakers is difficult, let alone the impact of the paleoscience content on its own. We do recall, however, that paleo aspects of**

the WGI report were well received by governmental delegates. In one case, information that had been relegated to a footnote was promoted to a bulleted point in response to interest in the topic by one or more governmental representatives.

I'd also be interested to know how the amount of paleo information across the whole report differed in AR6 compared with AR5. This would be easy to calculate for AR5, but would require a bit of work for AR6, to add up the content across multiple chapters, although the "paleo contents page" in Chapter 2, and/or the PAGES (2021) article would be helpful for this.

**Tallying the amount of paleo information across the whole report would be onerous. We focused on the summary documents (SPM and TS) because they are the two most widely read components of the report and therefore directly attest to the issue of "visibility and relevance" that the PAGES communication targeted. We will, however, include the analysis of citations to papers in *Climate of the Past*, which can be compared across the WGI reports of AR5 and AR6.**

On this note, line 32-34: PAGES (2021) is given as a reference for "paleoscience information was covered more comprehensively in AR6-WGI than in previous IPCC reports", but that statement is not evidenced in that PAGES article, which is just a series of links to the AR6 paleo sections.

**We view this extensive list of topics included in AR6 as evidence of the scope of coverage. For clarity, we will state that it is our opinion that the coverage was greater in AR6 and will add, "...as evidenced by the breadth of topics that considered paleoscience information." We also note in our piece that broader coverage is expected considering developments in the field. For the SPM, we name the additional findings that were included in AR6 that were not in previous SPMs.**

Line 35-36: It is an interesting claim that having a separate chapter stimulated more paleo research – is there any evidence for this?

**This is our conjecture. One of the informal reviewers of an earlier version of this piece also made this point. Considering the frequency to which the IPCC reports are cited in the primary literature, it seems like a reasonable statement. In our experience, policy-relevant questions facing knowledge gaps do stimulate research for all fields of science.**

Lines 59-62: Link to the Appendix which evidences this.
**Will do.**

Line 85: Link to the Appendix which gives the exact numbers of FAQs.
**Will do.**

(3) Overview of online comments (as of 19/7/2024)
The online comments and review provide a range of views and many interesting and important points, and I would encourage the authors to incorporate these into their revised paper.

**We address each of the online Community Comments as individual replies within this open discussion.**

CC2: It is an interesting suggestion to include BOTH a paleoclimate chapter and integration into individual chapters. However, I feel that the integration requires paleo Lead Authors in individual chapters, and if you add to this the paleo chapter Lead Authors, this will be unfeasible in terms of the number of paleo scientists in the report compared to other disciplines.
CC6: I fully agree with all aspects of this comment, which I think supports my own independent review above.
RC1: I also fully agree with the comments in this review. However, seeing that it was another paleo scientist involved directly in AR6, and also a modeler like me, I am slightly uncomfortable with the potential biases of the reviewers; however, I think that the wide engagement of the community in commenting on the article does go some way to counter that bias.

==Community Comment 1, Isabel Cacho==

**Make paleoscience useful to a wider community**

Referee's verbatim comments are in plain font, followed by authors' replies in **bold**.

I use the IPCC figures a lot in my undergraduate and post-graduate lectures. Figures from IPCC AR6 integrating proxy records, observations and projections have been extremely useful in my lectures and they have provided a strong basis for showing my students the value of the paleo-research. I agree that the AR6 approach has increased the value of the paleosciences to a wider climate community, which has often put us in a separate box.

**Thank you for your comment. We are delighted that the integration of paleo data along with other lines of evidence presented in AR6 is useful in your teaching. We agree that embedding paleoscience within multiple chapters exposes the subject and its relevance to a wider community of climate scientists.**

==Community Comment 2, Elsa Arellano-Torres==

**Include both a dedicated chapter and its integration into other areas**

Referee's verbatim comments are in plain font, followed by authors' replies in **bold**.

I believe that the evidence provided by paleoscientists must include both a dedicated chapter and its integration into other scientific areas. The perspective provided by the paleoclimate community allows for a much broader vision of how the climate system and intensive anthropogenic activity come together to shape current climate change. If the chapter dedicated to paleoclimates is eliminated, we may lose the context of paleoclimatic history. Humans soon forget the past because it is part of our adaptation to moving forward. However, avoiding reviewing history has led us to repeat it, making us more vulnerable and less aware of our mistakes. Thank you for making this space available.

**Thank you for your comment reaffirming the value of paleoscience. We agree with Referee 2 that both a dedicated chapter and its integration into other scientific areas would be difficult to achieve in practice and we will add a paragraph to the revised manuscript to explain this. Considering all aspects of the rapidly expanding field of climate science, space and expertise representing any climate science topic is exceedingly limited in IPCC reports. Given the constraints on the overall number of IPCC Lead Authors and the multiple areas of expertise that must be represented, it would be very difficult to have a sufficient number of paleoscience experts within both a dedicated chapter and disseminated across other chapters. Moreover, this would increase the challenges to ensure consistency and complementarity among chapters and reports, and to avoid gaps. We believe that paleoscience expertise in AR7 would be most effectively deployed where it leads to integration of paleoscience knowledge and demonstration of its relevance. This includes the WGII report where information from paleoscience is needed for a comprehensive assessment of issues related to adaptation and risk management.**

==**Community Comment 3, Michael Prather**==

**Other disciplines have increased their impact through integration**

Referee's verbatim comments are in plain font, followed by authors' replies in **bold**.

I strongly concur with Kaufman and Masson-Delmotte's Communication. For one, there are just too many disciplines in Earth system observations, analysis, and projections for each to have their own chapter in an IPCC Assessment Report. As it is, the reports are becoming too extensive and difficult to organize, maintain, pay for, and even get through governmental approval. IPCC is not maintained by the governments to write a text book, although I happily acknowledge that I have used IPCC chapters for my classes since 1995. For two, the IPCC process is not intended to tout a discipline, but to show how the scientific community can combine knowledge to address urgent societal needs, such as the upcoming Special Report on Climate Change and Cities.

The second reasoning follows my experience with the IPCC Atmospheric Chemistry chapters. For FAR1.5, SAR, and TAR, I led or co-led the atmospheric chemistry chapter. That period was one of rapid growth in the field and those chapters pushed the community

to answer some of the key questions.  In AR4 the atmospheric chemistry chapter disappeared, being absorbed into an amalgam 'Changes in Atmospheric Constituents and in Radiative Forcing.'  In AR5, it appeared in both the RF and Near-Term Climate Change chapters.  The chemistry community's publications were being used to address larger issues.  The best success of this integration was with Szopa and Naik's AR6 chapter on Short-lived Climate Forcers, which was requested in part by the governments to address specifically air quality in a changing climate.  While I still miss the luxury of having a dedicated atmospheric chemistry chapter in IPCC, I now believe that our community has made a greater impact with the application of the core science to more relevant needs.

**Thank you for your comment. We agree that integration of information is amongst the benefits of the IPCC process. The history that you recount of how the subject of atmospheric chemistry was treated in each of the assessment reports is a valuable perspective for the paleoscience community.**

Community Comment 4, Antje Voelker

**Multidisciplinary chapters strengthen interactions among scientific communities**

Referee's verbatim comments are in plain font, followed by authors' replies in **bold**.

As someone who acted as expert reviewer for several chapters of the Second Order Draft of the AR6 WGI report, I fully concur with the authors' views. As reviewer I greatly appreciated the efforts made by the authors who wrote those AR6 WG I chapters and bridged between chapters/topics and contributed to the FAQs. The integration of paleoclimatic and/or paleobiological evidence into the respective subject chapters was well done and set that evidence into the context of the recent day and future climate evolution. Being able to attend online workshops and webinars in disciplines focusing on modern ocean conditions and related best practices during the pandemic years, reiterated to me once more the different timescales we are studying and thus the different "languages" we often speak – and subsequently the still existing lack of interaction between disciplines (e.g., within the marine biodiversity field). Bridging gaps in understanding (and viewpoints) will only move forward if one (and I still include myself here) has to read the science and language of the different groups and understand their evidence. The multidisciplinary AR6 chapters did exactly that! I strongly believe that the integration of the "different timescales and languages" for a  specific topic is promoting paleoscience much more to the wider scientific community and to government agencies than a dedicated paleoclimate chapter, over which one could easily jump if the "language is too difficult" or the topic not of major interest.

**Thank you for your comment. We are pleased to know that you find the multidisciplinary chapters to be useful and that you agree that such integration of paleoscience information is the more impactful approach in the IPCC context. In our experience, the integration of knowledge across fields of climate science is facilitated**

**by the IPCC assessment process, which strengthens interactions among scientists with complementary expertise. This integration is promoted by distributing paleoscience expertise across the chapters and across the Working Groups. We will further emphasize this point in the revised manuscript.**

Community Comment 5, Celia Martin-Puertas

**Integration facilitates broader use of paleoscience**

Referee's verbatim comments are in plain font, followed by authors' replies in **bold**.

I would like to thank the authors for writing this rapid communication, which hopefully awakens interest to make an additional effort to better communicate and integrate palaeoclimate research into other climate and environmental disciplines. I see the point of a dedicated chapter as safeguard for relevancy and visibility, but I believe it should not be the final goal of including palae-evidence in the IPCC ARs. Combining palaeo with other evidence throughout the three WG reports is, however, a more effective way to both show how applicable and important palaeoclimate is and to motivate other climate science communities to use palaeo as a resource – it includes a variety of formats such as data, publications, briefings or co-production approaches.

As a suggestion, I would call the palaeoclimate community to be aware of the key policy-relevant topics the AR7 will target at. Delivering research outputs that deals with societally-relevant issues in the next years will help increase the number of palaeoscience citations in the next AR7 and promote impact through paleoclimate research.

**Thank you for your comment in support of our view, and for your suggestion to highlight some of the key policy-relevant topics in the IPCC report. We will mention some topics that we view as important for stronger inclusion in AR7 within the "opportunities" section of our piece. In addition, an index of paleo topics included in the AR6-WGI report is available at https://pastglobalchanges.org/news/quick-guide-paleoclimate-ipcc-ar6-2021-report.**

Community Comment 6, Eric Wolff

**Promote community engagement in the IPCC process**

Referee's verbatim comments are in plain font, followed by authors' replies in **bold**.

This is not really a scientific paper (even as a rapid communication) but an opinion piece. While I was always opposed to having different types of paper in CP, I think it would be better if this paper was headed "Opinion piece" as part of its title so that its status was entirely clear. It feels anyway a little strange to frame it as a rebuttal of what was a single sentence in an otherwise innocuous and ephemeral letter on the PAGES website.

Despite my misgivings expressed in the first paragraph, this is an issue of interest to the palaeoclimate community, and I am pleased that CP was chosen as the vehicle to discuss it. However for fairness and to make it a more objective article (as also expressed in the insightful comment by Brierley), it should either (a) start by expressing clearly the arguments both for and against a dedicated chapter, and then explain why the latter has been chosen, or (b) be part of two companion articles that present the arguments for and against. The present situation where this article acts as a rebuttal of an alternative position that has not been expressed (the arguments for the alternative are not presented in the PAGES letter) is rather unsatisfactory.

As an aside, the purpose of the discussion in CPD/EGUSphere is to discuss the merits of publishing an article, not to upvote or downvote an opinion!

The main benefit of publishing this article would be to clearly explain to the palaeo community how to engage with and get the attention of the IPCC process so that important palaeo insights make it into the report wherever needed. In my experience the review process is very important in this process, and is where any of us can make sure relevant literature is considered, that it is cited accurately and that the palaeo messages are consistent between chapters.

So, in conclusion, I am content that this could be published. However I think there should be clarity that it is not really a peer-reviewed scientific article, which should be reflected in the title; and that the paper should be more even-handed in explaining what was gained and lost when the palaeo chapter disappeared.

For clarity I myself agree that it is better to have palaeo distributed into chapters answering science and policy questions, but we need to be really careful there is enough palaeo expertise in each chapter team.

**Thank you for your comment and for your recognition of the value of publishing this article in *Climate of the Past*. It is indeed an opinion piece. We submitted it as a "Rapid Communication" because among the goals of this new type of manuscript is to "discusses matters of policy and perspectives related to the science of the journal" (https://www.climate-of-the-past.net/about/news_and_press/2024-06-11_launch-of-rapid-communications-as-part-of-cps-20th-anniversary-celebrations.html). We're happy to include "opinion piece" or similar phrase as part of the title, but we will defer to the CP editors and Copernicus Publications to make that decision. Either way, we will add "in this opinion piece, we…" to the abstract of the manuscript so that it is stated upfront.**

**We agree with your statement about the need for paleo expertise in each chapter team. We say that the success of the distributed approach "depends on engagement and substantial input from paleoscientists during the scoping phase of the report so the full breadth of relevant paleoscience topics is explicitly identified and effectively parsed among chapters, and key expertise is ensured within the selection of author**

teams." We will also add a link to the IPCC National Focal Points and Observer Organizations where individuals and organizations can contact representatives for input to the process. Furthermore, we will remind readers of your point about the importance of community engagement in the review process to make sure that new knowledge developments are included where relevant. We note that collective reviews of IPCC reports by early career scientists was especially fruitful in AR6 and could be strengthened for future reports (Moreno-Ibáñez et al., 2024; doi: 10.3389/fclim.2024.1395040).**

**In response to your comment and to those of the two Referees' and other Community Commentators, we will expand our discussion of the opposing viewpoint by adding a section on arguments favoring a separate section. We will also address the suggestion that paleoscience information be featured in both a dedicated chapter and distributed in other chapters.**

Community Comment 12, Nerile Abram

**The purpose and make-up of IPCC reports**

Referee's verbatim comments are in plain font, followed by authors' replies in **bold**.

Thank you to the authors for writing this opinion piece, and for sparking an important discussion for the paleoclimate community.

My own impression is that AR6 was a major step forward for the communication of climate change science to people outside of our scientific fields. This particularly came through in the very clear and impactful headline statements and excellent graphics in the Summary for Policymakers and Technical Summary that summarised key messages of the report. This type of presentation of our science is critical if we are to make our work accessible to the people who have the ability to bring about the changes that are urgently needed to limit future climate change.

**We appreciate your insightful comments about the summary documents in AR6, based on your experience with SROCC. We wrote the piece to explain how findings from paleoscience were integrated with other lines of evidence in the AR6, which highlighted their relevance and as a result led to stronger visibility within the AR6 high-level summaries, thereby making them more accessible to a wide audience.**

The purpose of IPCC reports is to be a policy relevant assessment of our scientific understanding of climate change. They aren't meant to be a textbook for the benefit of the scientific community; we already have the academic publishing framework that serves this purpose and drives our scientific endevours forwards. Much of the criticism of the proposal to continue distributing paleoclimate evidence across the chapters of the next IPCC reports comes from the way that global temperature reconstructions of the Common Era were

covered and presented in the Summary for Policymakers and underlying AR6 report. I agree that there are still a lot of scientific details to study in this sphere, and that there are different ways that this could have been approached and presented in AR6 based on the available scientific literature (e.g. as described in Esper et al 2024). But I also think that these criticisms somewhat miss the point of why we produce IPCC reports. Would including additional context on uncertainties in temperature reconstructions of the last 2000 years have changed the key take away message that it is important for us to get across to non-scientists? I don't think that any of the alternate approaches mentioned in the discussion alter the headline message that "*human influence has warmed the climate at a rate that is unprecedented in at least the last 2,000 years*" [title of AR6 WG1 Figure SPM.1].

**We agree that how Common Era global temperature was depicted in AR6 does not change the key message that recent climate change is unusual, which is based on more than just the PAGES 2k global temperature reconstruction, and considers independent evidence from the cryosphere and biosphere, and from climate models, as they too are presented in AR6. The PAGES 2k depiction of Common Era global temperature was highlighted in the IPCC AR6 as the outcome of a major community effort, with more recent publications suggesting alternative ways to assess and display uncertainties on seasonal or hemispheric temperature reconstructions, which provide valuable inputs for AR7. We call for the paleoclimate research community to work together and develop collective, timely assessments of the state of knowledge - as done for instance for climate sensitivity through WCRP Grand Challenges - to inform IPCC reports. Your comment clearly highlights that several of the other Community Comments do not fully encompass the purpose of IPCC reports, and that they have missed the point of our piece, as well as the points made by over half of the Commenters in this open discussion.**

One aspect that I don't think has come through in the discussion of this opinion piece so far is the way that integrating paleoscience into the chapters of IPCC reports assists in assigning calibrated language to key findings of the IPCC assessment process. Assigning confidence involves bringing together multiple lines of evidence so that together the amount of evidence and agreement across different lines of evidence is used to assign a calibrated level of confidence to the various aspects of climate change. By making sure that paleoclimate perspectives are part of these lines of evidence we strengthen the IPCC assessment process. My own experiences in being an author on the Ocean and Cryosphere special report (SROCC) was that without having paleoclimate experts embedded within each chapter it is very difficult to make sure that paleoclimate evidence is included as part of this assessment at all of the places where it can make a difference to the assessment that we put forward to policymakers. Abram et al 2020, QSR, demonstrates an example of this from SROCC.

**Thank you for your suggestion. Our revised manuscript will include the importance of integrating paleoclimate evidence to strengthen confidence in key findings. We share the perspective that distributing the contribution of experts in paleoscience across various chapters is important to ensure that available knowledge is assessed**

**wherever relevant. We agree that the paleoclimate community can strengthen IPCC reports by timely review publications addressing methodological aspects to best quantify past climate variations and associated uncertainties, allowing IPCC authors to focus on the integration of these findings with other lines of evidence for policy-relevant issues.**

The discussion around whether paleoclimate information should be assessed as a separate chapter in the next IPCC report, or be distributed across chapters, has noted that the option of having both would not be feasible within the IPCC framework. However, another possibility that could be proposed comes from the SROCC example of having an Integrative Cross-Chapter Box. This was different from other examples of IPCC cross chapter boxes in that it was part of the approved outline and had a dedicated number of pages in the report (although it didn't have a separate author team and authors were instead drawn from across the other main chapters). In the case of SROCC the Integrative Cross-Chapter Box was part of the approved outline for the report as it covered a topic that was deemed to be of extreme policy relevance (low-lying islands and coasts). I am not sure that a similar case could be successfully made for paleoclimate evidence. However, there are increased efforts to bring non-traditional lines of evidence into the IPCC process, particularly in relation to Indigenous knowledge. Perhaps delegates might approve an Integrative Cross Chapter Box on long-term perspectives on climate change for AR7 that would draw together paleoclimate perspectives, early historical accounts and Indigenous knowledge. If something like this were approved in the AR7 outline, then it would give added weight to ensuring that the author teams for the AR7 chapters included the expertise needed to bring these non-traditional lines of evidence to the assessment.

**We thank you for highlighting the importance of cross-chapter boxes and also cross-chapter papers as used in AR6-WGII for specific geographical syntheses. As an example, our piece highlights one cross-chapter box that features the role of "paleoclimate reference periods" and points to where information about each is included in the report. Building cross-chapter boxes into the approved outline for AR7 is an excellent suggestion to ensure visibility and coordination for specific topics and could help ensure inclusion of paleoscience expertise in AR7. We will add some information to the revised manuscript explaining how to have input to the IPCC process.**

Ultimately, ensuring that paleoclimate perspectives are part of the AR7 assessment will depend upon the paleoclimate scientific community producing clear and compelling publications that demonstrate how paleoscience adds to our understanding of climate change in ways that are policy relevant. One of the recent advances that has occurred in scientific publishing (and that helps to bridge the long time gaps between IPCC reports) is the publication of annual updates of the indicators of climate change (Forster et al ESSD) and the global carbon budget (Friedlingstein et al ESSD). There are also various special issues that annually publish summaries of the state of the climate (e.g. Climate Chronicles). The paleoclimate research community could develop a similar approach that summarises each year the advances that have been made in paleoclimate perspectives on climate change. If

an annual peer-reviewed publication of this type were set up to have sections that align with the approved chapter outline of AR7 then it would prove to be a very valuable resource for making sure that our science contributes deeply and comprehensively to the next IPCC assessment. And beyond an IPCC focus, I think that this type of publication would also serve to make paleoclimate science more assessable to researchers in other areas of climate science. Climate of the Past would be an ideal venue for this type of publication.

**The annual updates of key indicators and forcings of climate change are great examples of community-generated products in support of upcoming IPCC reports. The timeseries for each of these indicators can also be extended back in time and their values for well-studied paleoclimate reference periods can be constrained using evidence from paleo records. While "updates" to paleo records don't involve real-time monitoring, our reconstructions are extended and fortified by new paleo datasets. Generating publications that feature new information and that compile regularized datasets about key indicators and forcings as evidenced by paleo datasets and that align with the approved chapter outlines for AR7 is an excellent suggestion, one that we will include in our revised manuscript.**

References:
Nerilie J. Abram, Jessica A. Hargreaves, Nicky M. Wright, Kaustubh Thirumalai, Caroline C. Ummenhofer, Matthew H. England (2020) Palaeoclimate perspectives on the Indian Ocean Dipole, Quaternary Science Reviews, 237, 106302, https://doi.org/10.1016/j.quascirev.2020.106302
Esper, J., Smerdon, J.E., Anchukaitis, K.J. *et al.* The IPCC's reductive Common Era temperature history. *Commun Earth Environ* 5, 222 (2024). https://doi.org/10.1038/s43247-024-01371-1
Forster, P. M., et al., Indicators of Global Climate Change 2023: annual update of key indicators of the state of the climate system and human influence, Earth Syst. Sci. Data, 16, 2625–2658, https://doi.org/10.5194/essd-16-2625-2024, 2024.
Friedlingstein, P., et al., Global Carbon Budget 2023, Earth Syst. Sci. Data, 15, 5301–5369, https://doi.org/10.5194/essd-15-5301-2023, 2023.
Climate Chronicles. Nature Reviews Earth and Environment. https://www.nature.com/articles/s43017-024-00553-x

**Response to Community Comments 7-11: Michael Sigl, Rob Wilson, Jason Smerdon, Kevin Anchukaitis, and Kathryn Allen**

**There's more to IPCC paleoscience than Common Era climate reconstructions**

**Community Commenters CC7 through CC11 specialize in the climate of the Common Era. They are all co-authors of a recent perspective piece (Esper et al., 2024) that criticizes the PAGES 2k multi-method ensemble reconstruction of global mean surface temperature, which was featured in the 2021 AR6. Their comments largely overlap, so we respond to them together here. And, in response to their comments and**

others calling for a more balanced presentation, we will add a section to our text focusing on arguments favoring a separate paleoscience chapter.

The Commenters contend that a separate chapter is needed to ensure the quality and accuracy of the assessment of the state of knowledge. They focus exclusively on uncertainties associated with the global temperature reconstruction for the past 2000 years and the PAGES 2k reconstruction. The methodologies associated with temperature reconstructions of the past millennium was indeed a major focus of the AR4 and AR5 assessments, which helps explain the multi-method compilation of reconstructions adopted by PAGES 2k, but this is only one amongst an increasing number of aspects for which paleoclimate knowledge is addressed in IPCC reports.

We agree with Community Comment 12 that these Commenters missed the point of our piece, as well as the points made by over half of the Commenters in this open discussion. Our piece concerns the full breadth of paleoscience and how the subject in its entirety can be made more visible and societally relevant in context of IPCC reports. It was written in response to the statement made by the PAGES International Program Office, which was widely distributed to its >5000 subscribers from 125 countries, saying that "the visibility and relevance of paleoscience suffered" in AR6. Our revised manuscript will keep its focus on this perspective and on opportunities to expand the paleoscience content of future reports, and not the strengths and weaknesses of the PAGES 2k reconstruction. We also agree with Community Comment 12 that how Common Era global temperature was depicted in AR6 does not change the key message that recent climate change is unusual, which is based on far more than just the PAGES 2k global temperature reconstruction.

Nonetheless, as suggested, our revised manuscript will cite the Esper et al. (2024) perspective piece, which called for a separate paleo-focused chapter in AR7. We note, however, that their paper failed to cite the rebuttal by Neukom et al. (2022; 10.1016/j.dendro.2022.125965), which specifically addresses the putative variance loss in the PAGES 2k global temperature reconstruction. Esper et al.'s figure 5 shows the variance of the PAGES 2k ensemble mean instead of the full ensemble, despite the demonstration by Neukom et al. that the variances in the PAGES 2k reconstruction are consistent with published Northern Hemisphere reconstructions and with climate model outputs.

The Commenters maintain that the success of paleoscience in IPCC reports should not be determined by its prominence, but by its quality. Of course the accuracy of the assessment of the state of knowledge in IPCC reports is paramount. It is upheld through an extensive open review process overseen by designated Review Editors. These subject-matter experts ensure that all substantive comments are addressed in a balanced and transparent way. In this regard, we note the apparent criticism of the review process by one of the Commenters who called out a review comment about the PAGES 2k reconstruction in the first-order draft. He quoted the comment and stated

that it was rejected by the authors but he did not mention the authors' response to the reviewer (https://www.ipcc.ch/report/ar6/wg1/downloads/drafts-and-reviews/). It explains the reasoning for focusing on global rather than hemispheric and regional-scale temperature reconstructions in Chapter 2. That explanation was accepted by the chapter Review Editors who are familiar with the scope and purpose of the chapter in context of the entire WGI report. With respect to quality assurance, in our experience and from our conversations with other IPCC authors, the content of the report is more thoroughly reviewed and heavily scrutinized than any single publication in peer-reviewed journals. The quality of the information in IPCC reports can also be attributed to the readily accessible data that underlie the major findings, which enables traceability and reproducibility.

In addition to quality, we, like several of the Community Commenters on this open discussion, place high value on "visibility and relevance" of paleoscience. These are the attributes specifically called out by the PAGES community-wide communication because they are essential to raising awareness of our science across a broader audience, an opportunity afforded by a widely distributed product for non-specialists like the IPCC reports. Visibility and relevance can be reasonably measured by textual analysis, a common approach in social sciences. Ours shows increased usage of paleoscience information across the WGI contribution to AR6, and especially in the SPM. It's telling that before he read our piece, Referee 2 also counted mentions of paleoclimate information in the SPM as a means of evaluating the effect of distributing paleoscience across different chapters. Similarly, Referee 1 counted citations to this journal as a means of evaluating the success of paleoscience in AR6. Both of these independent analyses agree with ours.

Besides quality, the Commenters argue that the success of paleoscience in IPCC reports should be determined by the "completeness" of its assessments. Thoroughness is indeed a core principle of IPCC assessments, but there are practical limits to what can be included. The study of past Earth system changes goes far beyond global Common Era climate reconstructions and IPCC authors must represent its full breadth. Considering the very tight constraint on the number of words available to convey the importance of paleoscience, we believe that the breadth of paleoscience information included in the report, especially as it contributes to a multi-evidence-based assessment of high-level findings, is a highly relevant measure of its "completeness" (avoiding key gaps where relevant paleoclimate knowledge would have been omitted). Also, as part of the effort to limit words, each assessment report picks up where the previous report left off, focusing on new findings and those not included in previous reports. Methods and uncertainties associated with hemispheric temperature reconstructions for the Common Era had already been extensively covered in AR5.

Some of the Commenters, including Community Comment 2, suggest that paleoscience information in AR7 should be included both in a dedicated chapter plus

distributed in other chapters. While we certainly favor more exposure for paleoscience, we concur with the realistic view of Community Comment 3: "there are just too many disciplines in Earth system observations, analysis, and projections for each to have their own chapter…" Likewise there will likely be too few paleoscience experts among the group of IPCC Lead Authors both to populate a separate chapter and to embed into other chapters where they are needed to ensure that the information is actually included, as pointed out by Referee 2. In our revised manuscript, we will add a paragraph to address this suggestion by explaining our view that paleoscience expertise in AR7 would be most effectively deployed where it leads to integration of paleoscience knowledge and demonstration of its relevance.

One of the Commenters argued for a separate chapter because it "offers an opportunity for much greater participation by the palaeo-science community." We agree that participation by the paleoscience community is key to the success of our science in the IPCC process (a point also made by Community Comment 6), but we doubt that a separate chapter increases the opportunity for participation. There are avenues for participation, which we will mention in our revised manuscript. Foremost among them is volunteering as an expert reviewer of the draft reports. In this regard, we note that none of these five Commenters are listed among the AR6-WGI expert reviewers (https://www.ipcc.ch/report/ar6/wg1/downloads/report/IPCC_AR6_WGI_AnnexX.pdf).

More importantly, a major point of our piece is that participation is needed from paleoclimate scientists to work proactively and possibly through internationally coordinated professional organizations "to identify what appraisals of major research advances are missing from the literature and to initiate coordinated efforts to fill these gaps in support of AR7." The deep dive into uncertainties associated with large-scale temperature reconstructions of the Common Era called for by these Commenters is unlikely to be done by a small number of IPCC paleoscience authors who must also represent the full breadth of relevant paleoscience topics and integrate this information within the context of topical chapters. An example of such a community-led effort in support of a key IPCC topic, which we will highlight in the revised manuscript, is that by WCRP for the grand challenge of understanding climate sensitivity (Sherwood et al., 2020; 10.1029/2019RG000678). In our revised manuscript, we will also note the important role played by Contributing Authors who work as content experts to help draft chapter text alongside Lead Authors. In Chapter 2 of AR6-WGI, for example, there were 22 paleoscientists who served as Contributing Authors from outside the WGI Lead Author team.

Like the PAGES IPO communication that motivated this piece, we too encourage paleoclimate scientists, including Common Era climate specialists, to support the IPCC process. Our piece suggests several key IPCC topics with potential for stronger inclusion of information in AR7. For the Common Era, collective efforts are needed to distill paleoscience information regionally. This includes, for example, high-resolution

reconstructions of hydroclimate and information on extreme events and climatic impact-drivers. A separate chapter will not automatically fill these needs unless the community works proactively to provide focused recommendations for the AR7. This includes, amongst others, how to improve the representation of relevant information involving Common Era climate in a way that it can be used alongside other evidence in support of actionable science, and how to better display recent warming, as measured by global mean surface temperature, in a long-term context in a way that the general public and decision-makers can easily understand. The underlying publications with these advances are needed in support of the AR7 assessment cycle, which involves three Working Groups and will unfold in the relatively short timeframe of the coming years.